# Developing a COVID-19 mortality risk prediction model when individual-level data are not available

Noam Barda [1,2,3], Dan Riesel [1], Amichay Akriv[1], Joseph Levy[1], Uriah Finkel[1], Gal Yona[4], Daniel Greenfeld[5], Shimon Sheiba[5], Jonathan Somer[5], Eitan Bachmat [3], Guy N. Rothblum[4], Uri Shalit[5], Doron Netzer[6], Ran Balicer[1,2] & Noa Dagan [1,2,3✉]

At the COVID-19 pandemic onset, when individual-level data of COVID-19 patients were not yet available, there was already a need for risk predictors to support prevention and treatment decisions. Here, we report a hybrid strategy to create such a predictor, combining the development of a baseline severe respiratory infection risk predictor and a post-processing method to calibrate the predictions to reported COVID-19 case-fatality rates. With the accumulation of a COVID-19 patient cohort, this predictor is validated to have good discrimination (area under the receiver-operating characteristics curve of 0.943) and calibration (markedly improved compared to that of the baseline predictor). At a 5% risk threshold, 15% of patients are marked as high-risk, achieving a sensitivity of 88%. We thus demonstrate that even at the onset of a pandemic, shrouded in epidemiologic fog of war, it is possible to provide a useful risk predictor, now widely used in a large healthcare organization.

[1] Clalit Research Institute, Innovation Division, Clalit Health Services, Toval 40, Ramat-Gan, Israel. [2] School of Public Health, Faculty of Health Sciences, Ben Gurion University of the Negev, Ben-Gurion blvd. 1, Be'er Sheva, Israel. [3] Department of Computer Science, Ben Gurion University of the Negev, Ben-Gurion blvd. 1, Be'er Sheva, Israel. [4] Department of Computer Science and Applied Mathematics, Weizmann Institute of Science, Herzl 234, Rehovot, Israel. [5] Faculty of Industrial Engineering and Management, Technion University, Haifa, Israel. [6] Community Medical Services Division, Clalit Health Services, Arlozorov 101, Tel Aviv, Israel. ✉email: noada@clalit.org.il

The global coronavirus disease 2019 (COVID-19) pandemic is challenging healthcare systems around the world[1,2]. The wide range of outcomes observed, ranging from sub-populations that are mainly asymptomatic to subpopulations with substantial fatality rates[3], calls for risk stratification.

Such risk stratification can help to better tailor efforts, both for prevention (i.e., home quarantine, social distancing) and for treatment of confirmed cases (i.e., hospitalization vs. community isolation). In the face of equipment, services, and personnel shortages, risk stratification can allow better use of existing resources and improved outcomes.

To define who is at high risk for severe disease, the Centers for Disease Control and Prevention (CDC) defined the following criteria, such that if any exist, the patient is considered high risk[4]: people 65 years and older, people who live in nursing homes, and people with at least one of the following conditions—chronic lung disease, serious heart conditions, severe obesity, diabetes, chronic kidney disease, liver disease, or people who are immunocompromised.

Unlike the binary classification to low- and high-risk groups that results from decision rules such as the ones put forth by the CDC, the medical community is long accustomed to use more granular and individualized evaluation of patients' risk, which is usually quantified using multivariable prediction models[5]. Because patient risk is multifactorial in nature, with many interactions between the various factors, such models are well suited to the task of risk evaluation. Training of these models requires individual-level data, which are usually available from retrospective electronic healthcare record data[6,7] or from datasets of cohorts that were collected for research purposes[8,9]. However, such individual-level data of COVID-19 patients were not available when most western countries started forming their strategy to deal with the rise in COVID-19 patients.

Many attempts to provide risk prediction models for COVID-19 were described in the first months of the COVID-19 pandemic. In April 2020, Wynants et al. reviewed[10] prediction models for COVID-19, and found three that were trained on a general population to predict hospital admission due to pneumonia (as a proxy for COVID-19 pneumonia), and ten other prognostic models that were trained on COVID-19 patients to predict an outcome of death, severe disease or the need for hospital admission. However, most of these models are not reported at the usual standard required for prediction models[11], and Wynants et al. summarized them as "poorly reported, at high risk of bias, and their reported performance is probably optimistic[10]."

In this paper, we propose a hybrid methodology, which allows the construction of a multivariable prediction model without access to individual-level data pertaining to the current pandemic. This methodology first uses a baseline model trained on the general population in order to provide a granular ranking (discrimination) of the risk for severe respiratory infection or sepsis, which were hypothesized to share a common physiologic tendency with severe COVID-19 infection. Then, a postprocessing multicalibration algorithm[12] is used to adjust the predictions to published aggregate epidemiological reports of COVID-19 case-fatality rates (CFRs) in various subpopulations.

We demonstrate how this methodology was used in practice to develop a model during the first weeks of the COVID-19 pandemic and resulted in a model with good discrimination and calibration (validated once sufficient individual-level data for COVID-19 became available). This model is currently deployed and used in a large healthcare organization for prevention, testing and treatment decisions.

## Results
**Baseline model**. The characteristics of the population used for the training and testing of the final baseline model are described in

Supplementary Table 1. Among this population of 1,050,000 Clalit Health Services' (CHS) members over the age of 10 years, 11,718 (1.1%) positive outcomes of severe respiratory infection or sepsis were recorded over a follow-up period of 1 year.

The contribution and effect of the features of the baseline model for prediction of its chosen outcome, as measured by SHaply Additive exPlanations (SHAP) scores, are presented in Fig. 1. Figure 1a presents the overall importance and effect of all variables in the model. The SHAP scores in this figure display the contribution of each feature value (low vs. high) for decreasing or increasing the prediction value assigned to each patient. The figure also shows the predictive contribution of missingness (gray points)—whereby a missing value in a feature (e.g., chloride) is a signal regarding the patient's risk. Figure 1b presents the odds ratios across different values of three selected variables[13]. Similar figures for the rest of the variables are included in Supplementary Fig. 1.

Performance of the baseline prediction model on the test set is detailed in Fig. 2. The area under the receiver-operating characteristics curve (AUROC) was 0.820 (95% confidence intervals (CI): 0.811–0.828), which indicates good discrimination. The calibration plot, which runs very close to the diagonal, shows excellent calibration.

**COVID-19 mortality model**. The recalibration procedure of the baseline model to the COVID-19 CFRs among different age and sex groups terminated successfully. The aggregated corrections for each group are detailed in Supplementary Table 2. It is evident that the larger corrections made were for the older age groups, where the risk had to be increased substantially.

The model was deployed for large-scale use in CHS, a payer-provider healthcare organization in Israel, around mid-March 2020. When a sufficiently large local population of confirmed COVID-19 patients accumulated over the following weeks, performance of the COVID-19 predictions was tested. The last date that was allowed for confirmed cases to be included in the analysis was 3 months prior to the extraction date, thus allowing a minimum follow up period of 13 weeks. The empirical cumulative distribution function for the time of death of all COVID-19 patients in CHS, adjusted for censoring and the competing risk of cure is shown in Supplementary Fig. 2. This figure indicates that nearly all deaths occur by 13 weeks (91 days), thus confirming that the chosen length of follow-up period is sufficient.

The COVID-19 patient population used for validation included a cohort of 4179 COVID-19 patients that were diagnosed in CHS until April 16, 2020, with 143 (3.4%) deaths recorded until July 16, 2020. At the end of the follow-up period, 11 patients (0.3%) were still hospitalized and positive for COVID-19. A population table for the COVID-19 patient population is detailed in Table 1. The table shows the much higher fatality rate among the chronically ill and those of advanced age.

Discriminatory performance of the COVID-19 predictions in general and at specific thresholds is detailed in Table 2. The overall AUROC of the COVID-19 predictions is 0.943 (95% CI 0.926–0.956). When considering an absolute risk of 10% as the threshold, a fraction of 8% (95% CI 7–9%) of the population is identified as having a high risk, and this high-risk group contains 71% (95% CI 64–79%) of the patients who eventually died (sensitivity). If a patient was identified as a high risk by this cutoff, his or her probability for death was 30% (95% CI 26–35%) (positive predictive value (PPV)). At a 5% risk threshold, a fraction of 15% (95% CI 14–16%) of the testing population is found to be at high risk, with a sensitivity of 88% (95% CI 83–93%) and PPV of 20% (95% CI 17–23%). The CDC high-risk

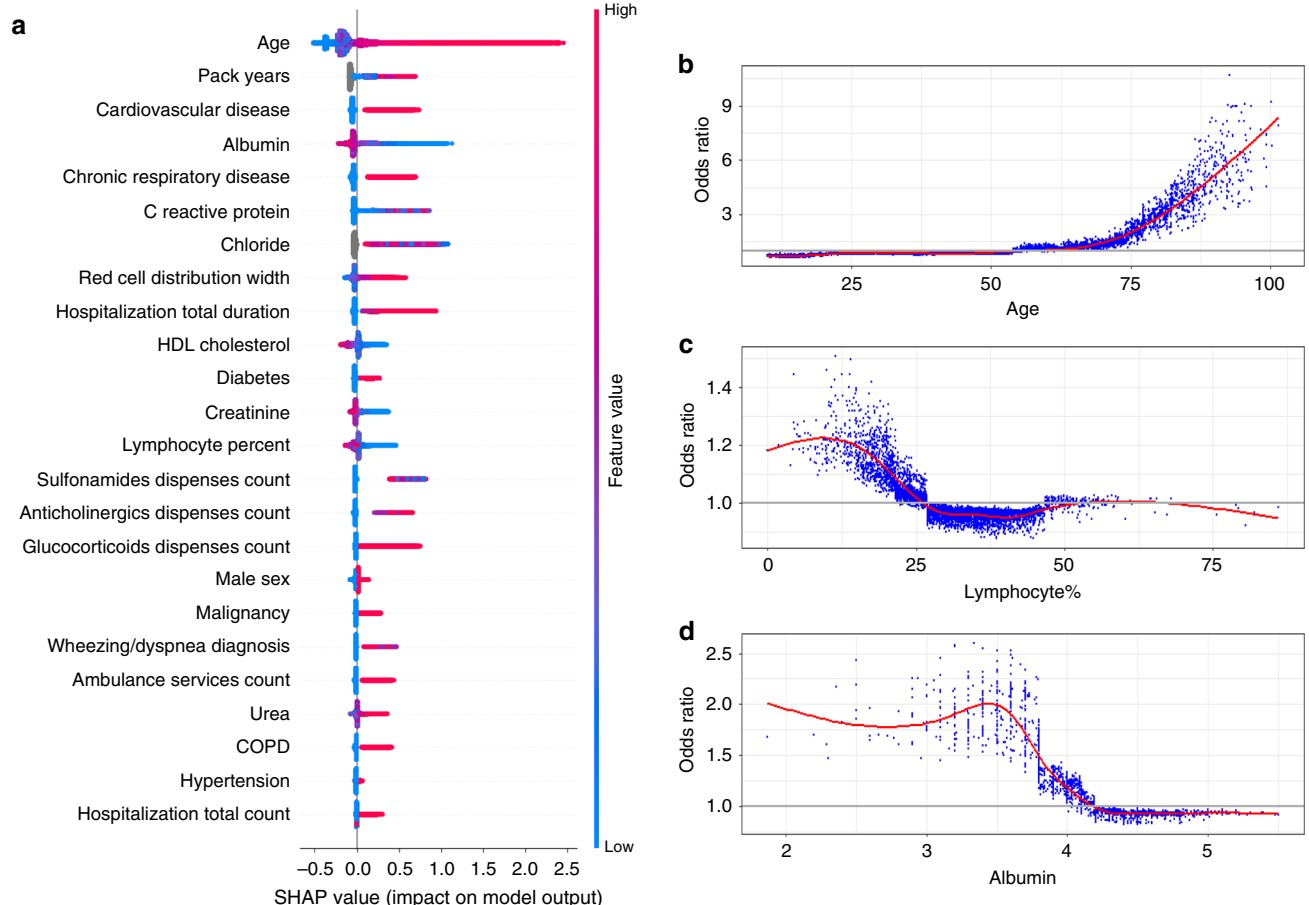

**Fig. 1 Summary and feature-specific SHAP values for the baseline model. a** A summary plot of the SHAP values for each feature. Going from top to bottom, features are ordered by their overall importance in creating the final prediction (sum of SHAP values). In each feature (line), every point is a specific case (individual), with colors ranging from red (high values of the predictor) to blue (low values of the predictor). Gray points signal missing values. The point's location on the X-axis represents the SHAP value—the effect the variable had on the prediction in this specific individual, with points further to the right marking that for that individual this covariate contributed to increasing of the risk and points to the left indicate that the covariate contributed to decreasing the risk. The vertical line in the middle represents no change in risk. **b** A plot of the odds ratio for different values of age. A smoothed red line is fit to the curve and a horizontal gray line is drawn at odds ratio = 1. **c** A plot of the odds ratio for different values of percent of lymphocytes in the blood. A smoothed red line is fit to the curve and a horizontal gray line is drawn at odds ratio = 1. **d** A plot of the odds ratio for different values of albumin. A smoothed red line is fit to the curve and a horizontal gray line is drawn at odds ratio = 1. **a** is based on the training set of the baseline population, n = 625,500 unique patients. **b–d** use a random sample of patients from this same population, n = 10,000 unique patients. SHAP SHapley Additive exPlanations, HDL high-density lipoprotein, COPD chronic obstructive pulmonary disease.

criteria identify 40% (95% CI 39–42%) of the same population as high risk with a sensitivity of 98% (95% CI 95–100%) and a PPV of 8% (95% CI 7–10%).

Figure 3 presents plots of the COVID-19 predictions' PPV against their sensitivity and of their sensitivity against the percent of patients identified as high risk, respectively. These plots also contain a graphical indication of the CDC criteria's corresponding values.

A calibration plot, comparing the ability of the COVID-19 predictions and the baseline predictions in accurately evaluating the absolute risk for COVID-19 mortality, is included in Fig. 4a. The figure shows that the baseline predictions were too low for the new outcome and that the recalibration procedure provided a marked improvement. The importance of the correction is further illustrated by decision curves (Fig. 4b). These curves, which compare the utility of the decisions that would have been made using the two models, show that decisions made using the COVID-19 predictions are superior to those made by the baseline model for the thresholds relevant for defining high-risk groups.

The sensitivity analysis for a composite outcome that considers both severe COVID-19 cases and COVID-19 mortality is presented in Supplementary Table 3. It can be noted that although the calibration adjustment was done for COVID-19 CFRs, the model also provides good discrimination for identifying those at risk for a severe course of disease. Specifically, the overall AUROC for the composite outcome is 0.901 (95% CI 0.882–0.920). At a 5% risk threshold, a fraction of 15% (95% CI 14–16%) of the testing population is found to be at high risk for the composite outcome, with a sensitivity of 73% (95% CI 67–79%) and PPV of 27% (95% CI 24–31%).

Access details to the code required to generate the baseline model predictions, as well as the data required for the adjustment and calibration of the baseline predictions to COVID-19 CFRs, are supplied under the code availability statement.

## Discussion

This work presents a model for prediction of COVID-19 mortality that was developed using a hybrid approach, by calibrating

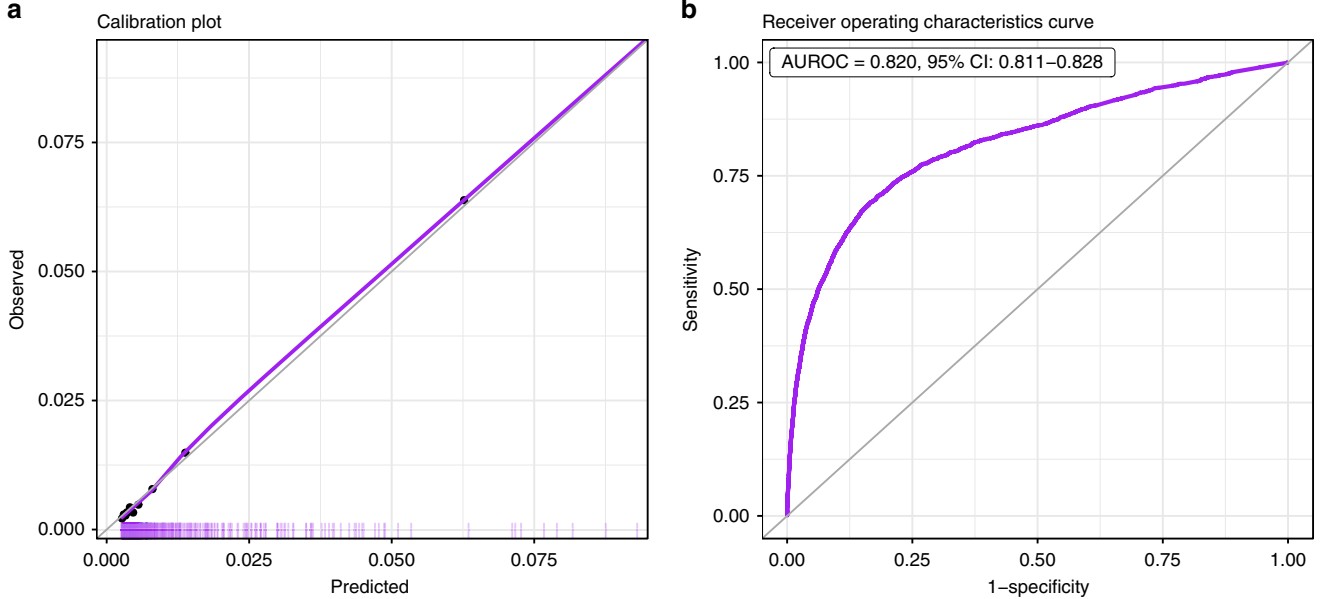

**Fig. 2 Performance charts for the baseline model. a** Calibration plot, plotting the observed outcome against the predicted probabilities. The diagonal gray line represents perfect calibration. A smoothed line is fit to the curve, and points are drawn to represent the averages in ten discretized bins. The rug under the plot illustrates the distribution of predictions. **b** Receiver-operating characteristics curve, plotting the sensitivity against one minus specificity for different values of the threshold. The diagonal gray line represents a model with no discrimination. The area under the curve, with its 95% confidence interval, is shown on the top-left. Both panels use the test population of the baseline model, $n = 315,000$ unique patients. AUROC area under the receiver-operating characteristics curve, CI confidence interval.

the predictions from a baseline model—designed to rank the population according to the risk for severe respiratory infections or sepsis—to reported COVID-19 mortality rates. The resulting model, developed rapidly under conditions of uncertainty, was used to evaluate the risk for COVID-19 mortality for the entire population of CHS members aged 10 years or older. It entered wide use in CHS around mid-March 2020, just before the first case of COVID-19 fatality in Israel. Since then, with the increase in the number of COVID-19 patients in Israel, some with severe outcomes, it became possible to validate its performance.

We showed that on a validation set of 4179 COVID-19 patients (3.4% death rate), the model was highly discriminative, with an AUROC of 0.943. We also showed that the postprocessing multicalibration phase resulted in a large improvement to the calibration of the baseline model when evaluated on COVID-19 patients (Fig. 4a). The improved calibration also translated to improved net benefit across the risk thresholds relevant for defining high-risk groups (Fig. 4b).

Since its development, the model has been used for many purposes within CHS. Its first use was for prevention purposes, with an intention to notify high-risk members of their increased risk for mortality should they get infected, to explain the importance of following social distancing instructions, and to provide information regarding telemedicine and other remotely accessible medical services. For that purpose, two cutoffs of high-risk groups were defined: the very high-risk group (absolute predicted risk ≥ 10%, about 4.3% of the CHS population over the age of 10 years) and the high-risk group (absolute predicted risk of 5–10%; 6.7% of the population). The first group received phone calls from their care providers to convey these messages personally, and both groups received a text message with similar information.

Risk stratification was also used for prioritization of COVID-19 RT-PCR tests, along with other criteria such as symptoms and potential exposure. Once tested for COVID-19, the model predictions were also used to decide on the place of treatment of

confirmed cases. This was done using the following methodology: first the treating physician makes a decision regarding the preferred place for treating a new patient, based on his or her clinical condition. If it is decided that the patient can be treated outside of a hospital, the physician checks the patient's risk group. Patients of the very high-risk group are reconsidered for hospital admission. Patients at the high-risk group can still be treated outside of a hospital, but their daily follow-up is more frequent than that of other patients.

The CDC and other authorities provided definitions of who should be considered to be at high risk[4]. These definitions work by stating a list of criteria, whereby patients are considered high risk if they fulfill any of them. There are two main issues with such binary classification. First, it does not allow the flexibility of choosing intervention cutoffs according to available resources or for interventions that are based on several risk levels. This is a problem especially when health resources are at a premium, for example when making personal phone calls by the medical staff. Second, defining a risk group by a list of risk factors (in the form of A or B or C etc.) usually results in many patients placed at the high-risk group, which is wasteful in its attempt to identify those who are truly at risk. This was noted in our COVID-19 patients' population, where the CDC criteria define 40% of the patients to be at high risk. While this group does include 98% of the death cases (sensitivity), a cutoff of 5% absolute risk by our more granular model allows defining only 15% of the COVID-19 patient population as high risk, while still attaining a sensitivity of 88%.

When first faced with the need to create a risk stratification model for COVID-19, there were no available individual-level data in Israel or in the public domain. The only information available that seemed sufficiently representative (i.e., not limited to inpatient populations) was that reported by the Chinese CDC[14], which reported CFRs for different subpopulations. Because these data cannot be translated directly to individualized

**Table 1 COVID-19 patient population characteristics table.**

| Variable[a] | Overall | Outcome = No | Outcome = Yes | Missing (%) |
|---|---|---|---|---|
| Overall | 4179 | 4036 | 143 | |
| Age, mean (SD) | 43.83 (21.18) | 42.54 (20.24) | 80.35 (12.92) | 0 |
| Age group (years) | | | | 0 |
| 10–19 | 433 (10.4) | 433 (10.7) | 0 (0.0) | |
| 20–29 | 942 (22.5) | 941 (23.3) | 1 (0.7) | |
| 30–39 | 654 (15.6) | 651 (16.1) | 3 (2.1) | |
| 40–49 | 550 (13.2) | 550 (13.6) | 0 (0.0) | |
| 50–59 | 512 (12.3) | 511 (12.7) | 1 (0.7) | |
| 60–69 | 526 (12.6) | 505 (12.5) | 21 (14.7) | |
| 70–79 | 284 (6.8) | 254 (6.3) | 30 (21.0) | |
| 80–89 | 186 (4.5) | 136 (3.4) | 50 (35.0) | |
| 90–99 | 92 (2.2) | 55 (1.4) | 37 (25.9) | |
| Sex | | | | 0 |
| M | 1959 (46.9) | 1884 (46.7) | 75 (52.4) | |
| F | 2220 (53.1) | 2152 (53.3) | 68 (47.6) | |
| Diabetes | | | | 0 |
| No | 3648 (87.3) | 3576 (88.6) | 72 (50.3) | |
| Yes | 531 (12.7) | 460 (11.4) | 71 (49.7) | |
| Hypertension | | | | 0 |
| No | 3435 (82.2) | 3394 (84.1) | 41 (28.7) | |
| Yes | 744 (17.8) | 642 (15.9) | 102 (71.3) | |
| Cardiovascular disease | | | | 0 |
| No | 3661 (87.6) | 3605 (89.3) | 56 (39.2) | |
| Yes | 518 (12.4) | 431 (10.7) | 87 (60.8) | |
| Malignancy | | | | 0 |
| No | 3915 (93.7) | 3809 (94.4) | 106 (74.1) | |
| Yes | 264 (6.3) | 227 (5.6) | 37 (25.9) | |
| Chronic respiratory disease | | | | 0 |
| No | 3818 (91.4) | 3698 (91.6) | 120 (83.9) | |
| Yes | 361 (8.6) | 338 (8.4) | 23 (16.1) | |
| Wheezing/Dyspnea diagnosis, Mean (SD) | 0.05 (0.45) | 0.04 (0.35) | 0.30 (1.50) | 0.0 |
| Albumin, mean (SD) | 4.13 (0.41) | 4.17 (0.38) | 3.63 (0.47) | 57.5 |
| Red cell distribution width, mean (SD) | 13.87 (1.48) | 13.80 (1.40) | 15.06 (2.13) | 48.9 |
| C-reactive peptide, mean (SD) | 1.12 (2.52) | 0.96 (2.15) | 3.46 (5.31) | 83.6 |
| Urea, mean (SD) | 33.52 (16.79) | 32.47 (15.04) | 51.14 (29.81) | 46 |
| Lymphocyte%, mean (SD) | 31.18 (9.18) | 31.38 (8.95) | 27.38 (12.27) | 39.8 |
| Chloride, mean (SD) | 103.71 (3.49) | 103.76 (3.42) | 103.18 (4.19) | 95.5 |
| Creatinine, mean (SD) | 0.82 (0.52) | 0.81 (0.50) | 1.07 (0.78) | 42.6 |
| High-density lipoprotein, mean (SD) | 49.47 (13.41) | 49.65 (13.32) | 46.47 (14.70) | 48.3 |
| Duration of hospitalizations, mean (SD) | 0.75 (6.03) | 0.62 (5.61) | 4.50 (12.77) | 0.0 |
| Count of hospitalizations, mean (SD) | 0.13 (0.59) | 0.11 (0.52) | 0.67 (1.45) | 0.0 |
| Count of ambulance rides, mean (SD) | 0.06 (0.39) | 0.05 (0.36) | 0.38 (0.79) | 0.0 |
| Count of sulfonamide dispenses, mean (SD) | 0.22 (1.46) | 0.17 (1.27) | 1.78 (3.77) | 0.0 |
| Count of anti-cholinergic dispenses, mean (SD) | 0.08 (0.90) | 0.06 (0.55) | 0.76 (3.83) | 0.0 |
| Count of glucocorticoid dispenses, mean (SD) | 0.17 (1.10) | 0.15 (0.97) | 0.65 (2.89) | 0.0 |

SD standard deviation.
[a]All variables were extracted at or before February 1, 2020, prior to the onset of the COVID-19 pandemic in Israel.

risk assessment, we chose to first train a predictor on a related outcome that was available in our retrospective data and then to adjust its predicted distributions to match those COVID-19 CFRs.

As the CFRs were the only available epidemiological evidence at the time of model development, it dictated that the outcome chosen was COVID-19 death, as opposed to the alternative composite of severe course of disease or death. Nonetheless, it was anticipated based on clinical considerations that this outcome will also provide a good ranking (discrimination) of the population with regard to the risk for a severe course of disease—those who will be ranked as having a high risk of death are likely also at high risk for the more general outcome of a severe course. This was indeed the case as can be noted from the results in Supplementary Table 3.

Interestingly, the multicalibration algorithm we used was borrowed from the domain of algorithmic fairness[15]; its intended use is to ensure calibrated and fair predictions for minority subpopulations that are underrepresented in a training set, and it is designed to run on hundreds of overlapping subpopulations (defined by many protected variables). In this case, however, the algorithm was used to make sure that the baseline model is calibrated for the reported CFRs of different age and sex groups. It should be noted that on top of the sex and age groups, we initially intended to calibrate the predictions to reported CFRs of populations with specific comorbidities, as any calibration target can be fed to the multicalibration algorithm in order to further refine the predictions' accuracy. This was eventually not done because of issues that were raised about the quality of the published comorbidity data in the Chinese report. As we could find

**Table 2 COVID-19 predictions performance table.**

| Model | Metric | Value |
|---|---|---|
| COVID-19 model | AUROC | 0.943, 95% CI: 0.926-0.956 |
| COVID-19 model 10% risk cut-off | Percent positive | 8%, 95% CI: 7–9% |
|  | Sensitivity | 71%, 95% CI: 64–79% |
|  | PPV | 30%, 95% CI: 26–35% |
| COVID-19 model 5% risk cut-off | Percent positive | 15%, 95% CI: 14–16% |
|  | Sensitivity | 88%, 95% CI: 83–93% |
|  | PPV | 20%, 95% CI: 17–23% |
| CDC high-risk criteria | Percent positive | 40%, 95% CI: 39–42% |
|  | Sensitivity | 98%, 95% CI: 95–100% |
|  | PPV | 8%, 95% CI: 7–10% |

*COVID-19* coronavirus disease 2019, *AUROC* area under the receiving operating characteristics curve, *CI* confidence interval, *PPV* positive predictive value, *CDC* Centers for Disease Control and prevention.

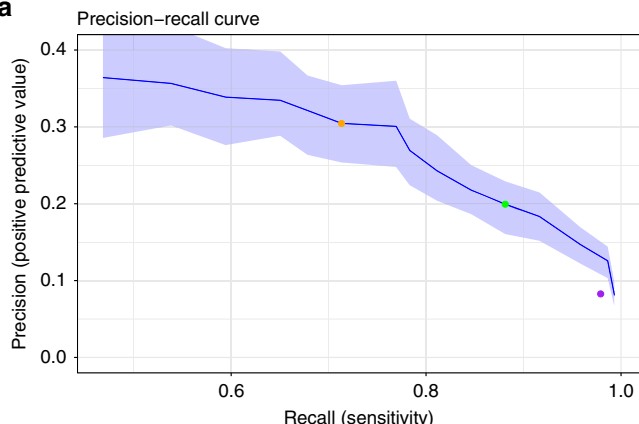

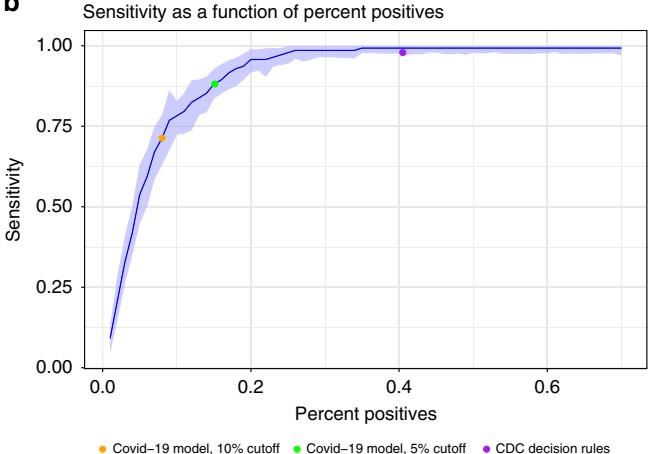

● Covid−19 model, 10% cutoff   ● Covid−19 model, 5% cutoff   ● CDC decision rules

**Fig. 3 Performance charts for the COVID-19 model. a** A plot of the positive predictive value against the sensitivity of the predictor for different thresholds. The central line represents the point estimates from the full population. The light band around the line represents point-wise 95% confidence intervals derived by bootstrapping. Only thresholds up to 15% absolute risk were plotted because of very low outcome rates in higher thresholds, which resulted in instability. The colored dots show the performance of three binary classifiers. **b** A plot of the sensitivity against the percent of patients identified as high risk for different thresholds. The central line represents the point estimates from the full population. The light band around the line represents point-wise 95% confidence intervals derived by bootstrapping. The colored dots show the performance of three binary classifiers. Both panels use the COVID-19 patient population, $n = 4179$ unique patients. CDC Centers for Disease Control and prevention, COVID-19 coronavirus disease 2019.

no other reports of CFRs for different comorbidity groups in representative COVID-19 populations (as of the time of model deployment), we opted to only adjust for age and sex.

We consider this approach to provide an efficient combination of individual-level data and epidemiologic reports. The resulting model can easily be further recalibrated whenever new data regarding different subpopulation CFRs become available, whether conditioned on a single characteristic (e.g., CFR in obese patients) or more (e.g., CFR in 50–70-year-old smokers).

Our baseline model identified relevant features from thousands of potential candidates. Among those, the model identified seven laboratory tests as being predictive of severe respiratory illness. Interestingly, most of these tests were later found to be indicative of severe COVID-19 infection. For example, recent reports highlight lymphopenia as a marker of COVID-19 disease severity[10,16–18]. C-reactive protein was also identified as an important feature for predicting a severe course of disease in hospitalized COVID-19 patients[10,17,19]. In addition, albumin, urea, and red cell distribution width were also mentioned as indicative of severe COVID-19 infection[19]. Lactate dehydrogenase, another variable that stood out as important is several studies[10,17–19], was not picked up by our baseline model. These findings of overlapping risk factors further strengthen the hypothesis that the baseline model correctly identifies a physiologic tendency and relevant susceptibility for a severe COVID-19 infection.

Three other models described in the literature employed a similar baseline model approach, with hospital admissions for respiratory disease used as a proxy for COVID-19 pneumonia. These models reported AUROC values of 0.73–0.81 (compared to 0.820 in our study)[10]. The models were all developed by DeCaprio et al.[20] on a cohort of ~1.8 million Medicare members. Contrary to the model we report, these models were not adjusted to the COVID-19 outcome and were not evaluated on COVID-19 patients.

In addition to these, there were ten other prognostic models for predicting death, severe disease or length of admission[10], which were trained on COVID-19 patients, almost exclusively from China. Wynants et al.[10] determined these studies to be at high risk of bias, due to a non-representative selection of control patients, exclusion of patients who had not experienced the event of interest by the end of the study, and high risk of model overfitting. In addition, they criticized most of the reports for low reporting quality. It should be emphasized that due to the urgent circumstances, many of these models were published as preprints, and have not yet been subjected to peer review. These studies were performed in an inpatient setting, with sample sizes ranging from dozens to several hundreds and a relatively high rate of severe outcomes (some with mortality rates of over 50%). Since

our work evaluates the risk for severe COVID-19 disease for the entire population, and is designed to be used prior to contracting the disease, a detailed comparison between the models is of less relevance.

Our work, which takes the hybrid approach of developing a population-based model that is then adjusted to COVID-19 mortality rates, has several strengths. It was conducted and reported according to the standard guidelines for prediction model reporting[11]. The baseline model was developed on a random and large sample of the CHS population, and validated on a separate test set, thus limiting the possibility of overfitting. In addition, the performance of the COVID-19 adjusted model was evaluated on all COVID-19 confirmed CHS cases that were diagnosed at least 3 months prior to the analysis. Given that Israel has a relatively high testing rate[21], and given that all tests and patients' status reports are collected centrally (whether they are

**a**

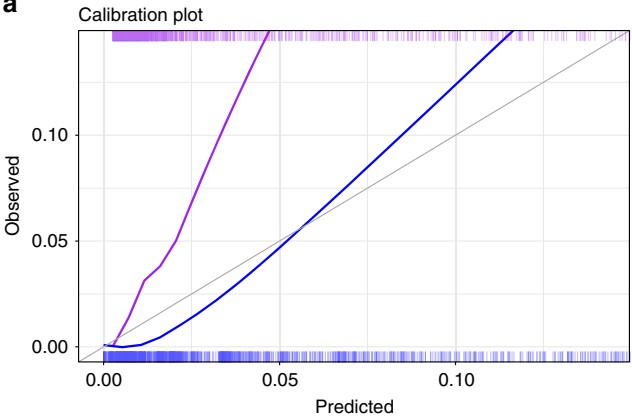

**b**

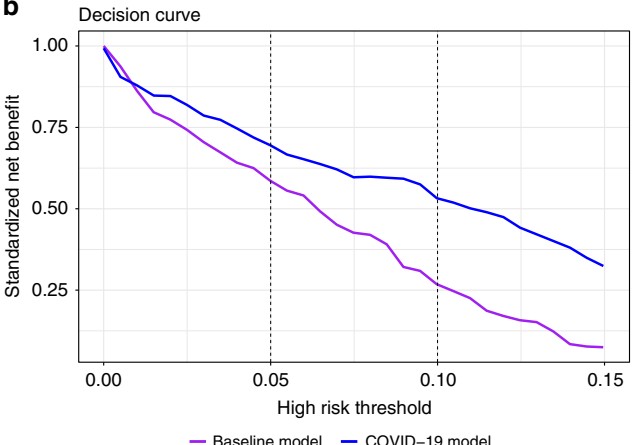

**Fig. 4 Calibration plot and decision curves comparing the COVID-19 and baseline models. a** Calibration plots plotting the observed outcome against the predicted probabilities of both models. The diagonal gray line represents perfect calibration. A smoothed line is fit to each curve. The rug above and under the plots illustrates the distribution of predictions for each model. The plot covers 95% of COVID-19 predictions. **b** The decision curve plots the standardized net benefit against different decision thresholds for both models. Net benefit is a measure of utility that calculates a weighted sum of true positives and false positives, weighted according to the threshold. Vertical dashed lines were added at relevant decision thresholds that were used in practice. Both panels use the COVID-19 patient population, $n = 4179$ unique patients. COVID-19 coronavirus disease 2019.

treated in the community or in the hospital), the possibility for selection or sampling bias is markedly reduced.

This work also has several limitations. First, the method described depends on being able to construct a discriminative model for a related outcome, for which the resulting ranking is relevant to the outcome of interest. This is usually not known accurately in advance, particularly for emerging diseases such as COVID-19. Second, the method requires subpopulation-specific CFRs that are relevant to the study population. These will usually be found in foreign populations and will need to be corrected to the local population. The correction method used in this study, using a linear probability model, is coarse, and could result in significant bias if the populations are sufficiently different. Third, patients that were diagnosed later in the follow-up period had a shorter follow-up time (with a minimum of 3 months). While this could potentially result in under-estimation of the outcome risk, a sensitivity analysis shows that this follow-up is sufficiently long to identify nearly all deaths.

Last and most important, without individual-level data, there is no way to test the resulting predictions. As a result, a poorly

performing model could find its way into clinical use at a critical time. This can be counteracted, at least partially, by evaluating the resulting model and predictions manually, using clinical domain knowledge, as was performed in the CHS prior to model deployment. This also emphasizes the importance of validating the model on individual-level data as soon as such data becomes available.

Application of the findings described in this paper can be viewed in two different manners. First, the COVID-19 predictor described can be potentially used directly (via the provided analysis code) if the required input data are available. As the overall COVID-19 epidemiology and risk factors witnessed in Israel were very similar to those described in other countries, it is likely that the predictions will provide good discrimination in other settings as well. Despite this, as in any application of an external prediction model, external validation should be performed prior to using the model in a different setting. In this scenario, it is important that the distribution of predictors is as similar as possible to the training population, so for variables that can be missing (e.g., laboratory results), in which the missingness is used in the model as an important input by itself, data should not be completed or imputed.

A second potential application of the described findings is the use of the general process that was applied in this work for creating predictions when individual-level data regarding the outcome of interest in not available. This methodology can be used whenever a model is desired for a disease and a specific health organization does not possess the data to train and validate it, but the infrastructure does exist to predict the risk for a related disease. This can be particularly useful, for example, in the early stages of dealing with an emerging or reemerging disease (infectious or otherwise), or when individual-level data does not exist for technical reasons. Specifically, the same methodology can be applied to translate a related prediction model into a COVID-19 predictor by healthcare organizations that do not yet have available COVID-19 outcome data (as was done in this paper).

To conclude, in this work we described the development and use of a prediction model for a novel infection, when individual-level data are not yet available. We found that even in the midst of a pandemic, shrouded in epidemiologic fog of war, it was possible to provide a useful prediction model for COVID-19 with good discrimination and calibration. It appears that the choice of a related outcome was able to provide an accurate ranking of the population, and the multicalibration approach was useful for integration of published epidemiological data into the model. We also demonstrated that this approach is more efficient in identifying high-risk individuals compared to a list of risk criteria, such as the one suggested by the CDC.

The methodology described here can be used in other populations lacking individual COVID-19 data and in future similar circumstances for other emerging diseases or situations in which an outcome of interest is not yet available in the local data. As a healthcare organization, we consider risk stratification tools as important for both proactive prevention measures and for care decisions regarding confirmed patients. This need becomes even more critical when health systems are facing extreme loads and there is a need to properly assign available resources.

## Methods
**Setting and source of data**. This is a retrospective cohort study based on the data warehouse of CHS, a large integrated payer-provider healthcare organization operating in Israel. Health insurance in Israel is mandatory and is offered by several payer-provider organizations that directly supply most of the medical services to their insured population and indirectly pay for additional required services. CHS, the largest of these organizations, insures over half of the Israeli population (~4.5 million members), which are a representative sample of the entire population. The CHS data warehouse contains both medical data (primary care, specialist care, laboratory data, in-network hospitalization data, imaging data etc.) and claims data

(mostly out-of-network hospitalization data). CHS has been digitalized since the year 2000 and has a low yearly drop-out rate (~1–2%), allowing long follow-up of patients in the data warehouse.

**Baseline model**. The entire CHS population over the age of 10 years was designated as relevant for the prediction. The exclusion of patients under the age of 10 years was done based on the very low rates of COVID-19 mortality described in this young population, in contrast to young children's susceptibility to severe respiratory infections by other pathogens (which may have been detected by the baseline model and would potentially create bias in the final COVID-19 model).

The index date was set to June 1, 2018. Baseline covariates were extracted in the year prior to this date, with the value set as the last result available. A positive outcome was defined as a hospitalization with diagnoses of pneumonia, other respiratory infections or sepsis, or a positive influenza PCR result (i.e., influenza infection that required inpatient care). Outcome data were collected over a follow-up period of 12 months (until May 31, 2019).

The baseline model was trained in two phases—initial training for feature selection and final training for creation of the baseline model. In the first phase, a population of 1,000,000 members, selected randomly from CHS members of the relevant age range, was used. This population was run through an automatic prediction pipeline to explore important features for the prediction task, out of all covariates available in the CHS data warehouse (~15,000 potential features).

Once this model was trained, the top 30 features by cumulative information gain (a measure from information theory)[22] were passed to a group of clinicians that chose a subset of features with meaningful and intuitive medical sense. In addition, variables describing medical conditions reported to be related to severe COVID-19 infection were added manually. This selection process, which resulted in 24 selected features, was performed to improve the interpretability of the resulting model and to allow its sharing with other providers. The full list of features used in the eventual model and the exact definition of the outcome used are detailed in Supplementary Table 4.

The final baseline model, with the selected features, was trained and tested using another random sample of 1,050,000 individuals (that were not included in the feature selection phase). This population was then randomly divided in a 70/30 ratio to training–validation and test sets, respectively (the training–validation was further divided to training and validation sets at a ratio of 5:1). Each individual was only included once in the analysis.

The final model employed was a decision-tree-based gradient boosting model using the LightGBM library[22] with default hyperparameters[23]. The validation set was used for early stopping, with AUROC used as the performance measure. Decision-tree models are capable of handling missing data without the need for imputation (assigning them to the branch of a split that minimizes the loss function), and accordingly, none was performed.

Contribution and effect of the selected features were calculated using SHAP scores[13]. SHAP scores are feature importance scores based on Shapley values from game theory. They measure the effect that a specific value in a specific covariate has for the prediction of a certain individual (so the same variable with a specific value could have a different SHAP value for different individuals, based on its interaction with other characteristics of that individual). In addition to the SHAP values themselves, we also used them to express the contribution of specific covariates to the predicted risk of each individual in terms of odds ratios (an explanation regrading this transformation is provided in Supplementary Methods 1).

The model was scored on the test set for metrics of discrimination and calibration. Discrimination was measured as the AUROC. Calibration was measured using a smooth calibration plot[24]. CI for the various performance measures were derived using the bootstrap percentile method[25] with 5000 bootstrap repetitions.

**Adjustment of the baseline model for COVID-19 mortality**. The final baseline model was applied to the entire CHS population over the age of 10 years as of an index date of February 1, 2020, which was chosen to avoid the effects of COVID-19 (the first COVID-19 case was identified in Israel on January 29, 2020). Aiming to eventually produce a model for the risk of mortality from COVID-19 among the infected, after producing individual risk assessment for what we hypothesized to be a physiologic tendency to severe COVID-19 infection, there was a need to recalibrate the results to the outcome of interest—mortality due to COVID-19 infection. This process was done using reported COVID-19 CFRs for different subpopulations, taken from a report by the Chinese center for disease control[14]. The data in this report were presented as one-way conditionals, i.e., the probability of mortality given being in a specific age range, or of a specific sex group.

Before applying the one-way conditionals from the Chinese population to the Israeli population, they were corrected to account for the different demographic distributions of the two populations. For this purpose, a linear probability model was trained on the Chinese data with 10-year age groups and sex as the predictors. This model has the advantage of only requiring the pair-wise covariance between the independent variables, which were available for the Chinese population as part of the global burden of disease study 2017[26,27]. Once this model was trained, the

corresponding conditionals were calculated on the Israeli population, using national statistics pertaining to its age and sex composition.

Adjustment of the predictions from the baseline model was done using a multicalibration algorithm[12]. This algorithm acts as a postprocessing phase for any baseline prediction model and works by iteratively adjusting the predictions of each subgroup (e.g., females aged 60–69) so that their mean is equal to the mean of the observed outcomes in that subgroup. The algorithm iterates until no subgroups remain whose average is further from their target value by more than a predetermined tolerance, which was set to 1%. Since the subpopulations are overlapping, this process requires several iterations until it converges (a detailed pseudocode of this algorithm is provided in Supplementary Methods 2). The predictions that were output from the recalibration procedure were used as the final model for predicting the risk of death from COVID-19 infection (i.e., the COVID-19 predictions).

See Supplementary Methods 3 for a complete technical description of the adjustment procedure.

**Testing the adjusted outcomes**. Once enough confirmed COVID-19 patients accumulated in Israel, the COVID-19 predictions were tested on the entire CHS COVID-19 patient population.

The study population was extracted on July 16, 2020, and included all patients diagnosed until April 16, at least 3 months prior to the extraction date (to allow sufficient time for the outcome to occur). To validate this decision, the empirical cumulative distribution function of time-to-death from COVID-19 was generated using all patients in CHS' database diagnosed with COVID-19 until the extraction date. This was derived accounting for censoring and the competing risk of cure, calculated using the Aalen-Johansen estimator[28].

Patients were diagnosed using RT-PCR tests from nasal and pharyngeal swabs performed between the beginning of the outbreak in Israel and the extraction date. The outcome used was death during a hospitalization following a new COVID-19 infection, as identified from national registries which are updated daily.

The COVID-19 predictions were scored for discrimination using the AUROC. To assess the success of the calibration adjustment procedure, calibration was compared to that of the baseline model using calibration curves (with both the baseline and the final COVID-19 models evaluated against the COVID-19 mortality outcome). The effects of the recalibration were also assessed via decision curves, which present net benefit against different decision thresholds and are used to evaluate the utility of decisions made based on the model[29].

In addition, plots of the PPV against the sensitivity (precision–recall curve), and of the sensitivity against the percent of patients identified as high risk were drawn across different thresholds. This was done to present the sensitivity and PPV that result from all possible shares of the population that could be defined to be at high risk. These plots were also used to demonstrate the comparative performance of the binary classification that results from the American CDC criteria for defining high-risk patients[4].

In addition to the main analysis that considered the model's ability to predict COVID-19 death risk, a sensitivity analysis was conducted to evaluate the model's ability to also provide good discrimination for a composite outcome that considers severe disease in addition to mortality. Israeli hospitals were instructed by the ministry of health to define disease severity according to the US National Institutes of Health's definition[30], with the status updated daily. A patient was defined as suffering from severe disease if defined severe at any point during the index hospitalization.

**Ethical approval**. This study was approved by CHS' institutional review board (0143-19-COM1, 0052-20-COM).

**Analysis**. Data were collected using SQL Server 2017. The baseline model development was done using Python version 3.6, Anaconda version 5.1.0, and LightGBM version 2.2.3. The model performance analysis and plot creation were done in R version 3.5.2.

**Reporting summary**. Further information on research design is available in the Nature Research Reporting Summary linked to this article.

## Data availability
Access to the data used for this study can be made available upon request, subject to an internal review by N.B. and N.D. to ensure that participant privacy is protected, and subject to completion of a data sharing agreement, approval from the institutional review board of CHS and institutional guidelines and in accordance with the current data sharing guidelines of CHS and Israeli law. Pending the aforementioned approvals, data sharing will be made in a secure setting, on a per-case-specific manner, as defined by the chief information security officer of CHS. Please submit such requests to N.D.

## Code availability
The analytic code required to produce the COVID-19 model predictions is available at: https://github.com/clalitresearch/COVID-19-Model.

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

## Acknowledgements

We want to thank Adi Berliner, Shay Ben-Shahar, Anna Kuperberg, Reut Ohana, Nurit Man, Irena Livshitz, Alon Schwartz, Nir Shahar, and Ilana Roitman for contributing to the model's implementation. We want to thank Mark Katz, Anna Kuperberg, Morton Leibowitz, and Oren Oster for helping with the outcome definition. We want to thank Shay Perchik, Michael Lischinsky, and Galit Shaham for helping with Quality Assurance. We want to thank Ilan Gofer for organizing the data. G.N.R. reports grants from Israel Science Foundation, grants from Israel-US Binational Science Foundation, grants from European Research Council, and grants from Amazon Research Award during the conduct of the study. U.S. reports personal fees from K-health, grants from Israel Science Foundation, and grants from Yad Hanadiv, outside the submitted work.

## Author contributions

N.B., D.R., R.B., and N.D. conceived and designed the study. N.B., D.R., A.A., J.L., U.F., and N.D. participated in data extraction and analysis. G.Y., D.G., S.S., J.S., G.N.R., and U.S. participated in the conception and creation of the model adjustment process. D.N. was responsible for the model use in practice. N.B. and N.D. wrote the manuscript. All authors critically reviewed the manuscript. R.B. and N.D. supervised the entire study process.

## Competing interests

The authors declare no competing interests.
