## [Peer Review File · Nature Communications]

REVIEWER COMMENTS

Reviewer #1 (Remarks to the Author):

I thank the authors for their efforts in improving the paper. I have some further comments for clarification.

- Title is actually not conform TRIPOD, it should mention that the study develops and validates a prediction model.
- The addition of 'by cumulative information gain' like that, without ref and short description is unclear to many
- You used default hyperparameters for the gradient boosting model. Why, and can you specify what the hyperparameters are?
- SHaply Additive Explanatations (on p6 I think): correct typos.
- The outcome: now the authors write "6 weeks prior to the extraction date to allow sufficient time for the outcome to occur" and in the next paragraph "death during a hospitalization following a new covid-19 infection". The latter is a clear definition, but I am wondering whether there were no cases who were still alive and hospitalized at May 30th? I would think there are, and this has to be mentioned.
- About the newly added sensitivity analysis:
 - a. this was done using the recalibrated predictions based on CFR, right?
 - b. Please provide a more specific definition of severe disease than "resources such as ventilators and ICU beds", including when the classification into severe or not was made
 - c. I am not a fan of this post hoc analysis (the project starts with a proxy outcome, recalibrates it for mortality, and then checks performance for yet another outcome), but I understand this was added in response to another reviewer.
- Figure 1: a better quality figure would be nice. Also, the figures on the right are interesting, do you have figures for all predictors in Supplementary? In the footnotes, the authors now write "red (high values)" and "blue (low values)". Are this high/low risk estimates? Please clarify.
- In the section "covid-19 mortality model", the authors mention 135 deaths recorded until April 18th. I am confused, I thought data was extracted on May 30th?
- Right after, the authors write "there were no losses of follow-up". I am not sure I understand, because data were extracted on May 30th. This relates to my question above: where there patients who were alive and hospitalized on May 30th?
- Hospitalization total duration: this variable has a different label in Table 1, but the scale is still missing. Also, I hope this does not refer to the hospitalization for covid-19. If so, that would make the whole model invalid. Please clarify.
- In the version with track changes, Figure 4 has not changed. The calibration and decision curve must surely look different now that more data was added. Please check that the correct versions

are used.

Ben Van Calster, KU Leuven, Belgium

Reviewer #2 (Remarks to the Author):

The authors have satisfactorily addressed all of my comments

Reviewer #1 (Remarks to the Author):

I thank the authors for their efforts in improving the paper. I have some further comments for clarification.

First, we wish to thank Professor Van Calster for his many insightful comments. We feel they have much improved the manuscript.

-Title is actually not conform TRIPOD, it should mention that the study develops and validates a prediction model.

We wish to thank the author for pointing this out. The manuscript's title was accordingly changed to *"Performing risk stratification for COVID-19 mortality when individual level data is not available – development and validation of a prediction model"*.

-The addition of 'by cumulative information gain' like that, without ref and short description is unclear to many

We understand that this concept cannot be presented without an appropriate reference. We added context and a suitable reference to the mention. The manuscript now reads: *"Once this model was trained, the top 30 features by cumulative information gain (a measure from information theory)¹³ were passed..."*

-You used default hyperparameters for the gradient boosting model. Why, and can you specify what the hyperparameters are?

We used default hyperparameters because, in our experience, they consistently provide good performance when used with a large population that has many outcomes. The full list of default hyperparameters is long, which prevents listing them in full in the manuscript. In order to address this point we added a reference to LightGBM's API to the manuscript, where the list of all parameters and their default value is specified (<https://lightgbm.readthedocs.io/en/latest/Parameters.html>):

"The final model employed was a decision-tree-based gradient boosting model using the LightGBM library¹³ with default hyperparameters^{14"}.

-SHaply Additive Explanations (on p6 I think): correct typos.

We wish to thank the reviewer for noticing this oversight; the typo was corrected, and now reads: *"SHaply Additive exPlanations (SHAP) scores"*.

-The outcome: now the authors write “6 weeks prior to the extraction date to allow sufficient time for the outcome to occur” and in the next paragraph “death during a hospitalization following a new covid-19 infection”. The latter is a clear definition, but I am wondering whether there were no cases who were still alive and hospitalized at May 30th? I would think there are, and this has to be mentioned.

We took the chance to once more rerun the analysis, and to provide the requested addition of hospitalization cases that remain at the end of the follow-up period. Extracting on July 16th 2020, we used cases at or before April 16th 2020, allowing for full 3 months (13 weeks) of follow-up. All the results, tables and figures were updated accordingly. Of the 4,179 patients in the study, 11 (0.3%) patients were still hospitalized at the extraction date and were not deemed recovered from COVID-19.

The relevant paragraphs now read:

Methods section:

"The study population was extracted on July 16th, 2020, and included all patients diagnosed until April 16th, at least 3 months prior to the extraction date (to allow sufficient time for the outcome to occur)."

Results section:

"The last date that was allowed for confirmed cases to be included in the analysis was 3 months prior to the extraction date, thus allowing a minimum follow up period of 13 weeks. The empirical cumulative distribution functions for the time-of-death of all COVID-19 patients in CHS, adjusted for censoring and the "competing risk" of cure, is shown in Supplementary Figure 3. This figure indicates that nearly all deaths occur by 13 weeks (91 days), thus confirming that the chosen length of follow-up period is sufficient".

"The COVID-19 patient population used for validation included a cohort of 4,179 COVID-19 patients that were diagnosed until April 16th 2020, with 143 (3.4%) deaths recorded until July 16th, 2020. At the end of the follow-up period, 11 patients (0.3%) were still hospitalized and positive for COVID-19".

-About the newly added sensitivity analysis :

a. this was done using the recalibrated predictions based on CFR, right?

Indeed, it was done using the same predictions recalibrated for mortality.

b. Please provide a more specific definition of severe disease than “resources such as ventilators and ICU beds”, including when the classification into severe or not was made

The paragraph was rewritten to better reflect the definition of severe disease. It now reads:
"In addition to the main analysis that considered the model's ability to predict COVID-19 death risk, a sensitivity analysis was conducted to evaluate the model's ability to also provide good discrimination for a composite outcome that considers severe disease in addition to mortality. Israeli hospitals were instructed by the ministry of health to define disease severity

according to the US National Institutes of Health's definition²³, with the status updated daily. A patient was defined as suffering from severe disease if defined severe at any point during the index hospitalization".

c. I am not a fan of this post hoc analysis (the project starts with a proxy outcome, recalibrates it for mortality, and then checks performance for yet another outcome), but I understand this was added in response to another reviewer.

The reviewer's comment is accurate. As mentioned, the sensitivity analysis was performed at the request of another reviewer.

-Figure 1: a better quality figure would be nice. Also, the figures on the right are interesting, do you have figures for all predictors in Supplementary? In the footnotes, the authors now write "red (high values)" and "blue (low values)". Are this high/low risk estimates? Please clarify.

As requested by the reviewer, we tried to create the figure in higher quality. If the quality is still not satisfying, we would be happy to provide the separate graphs that compose this figure in a vector graphics format (such as SVG).

In addition, a Supplementary Figure 2 was added with the plots for all the predictors in the baseline model.

Finally, regarding the colors in Figure 1A – these colors correspond to the values of the predictor. To clarify this, the text in the caption of the figure was altered to "*with colors ranging from red (high values of the predictor) to blue (low values of the predictor)*". We wish to thank the reviewer for pointing out that this point was not clear enough.

-In the section "covid-19 mortality model", the authors mention 135 deaths recorded until April 18th. I am confused, I thought data was extracted on May 30th?

The reviewer is correct in his observation, and indeed the previous version contained an error in the reference to the correct date. The error is now corrected and the paragraph reads: "*The COVID-19 patient population used for validation included a cohort of 4,179 COVID-19 patients that were diagnosed until April 16th 2020, with 143 (3.4%) deaths recorded until July 16th, 2020. At the end of the follow-up period, 11 patients (0.3%) were still hospitalized and positive for COVID-19*".

-Right after, the authors write "there were no losses of follow-up". I am not sure I understand, because data were extracted on May 30th. This relates to my question above: where there patients who were alive and hospitalized on May 30th?

Indeed, as requested above, we now added the count of patients that were still hospitalized and COVID-19 positive at the end of the follow-up period (11 patients, 0.3% of the sample).

-Hospitalization total duration: this variable has a different label in Table 1, but the scale is

still missing. Also, I hope this does not refer to the hospitalization for covid-19. If so, that would make the whole model invalid. Please clarify.

The name of the variable was corrected to "*Duration of hospitalizations*" in all tables. It is measured in days, as listed in the "units" column. As with the other variables, the extraction for the predictors was performed on February 1st, 2020, and considered a single year backwards. Accordingly, the methods section states that: "*Baseline covariates were extracted in the year prior to this date*", and we also made sure that the same time horizon for the extraction is mentioned in Supplementary Table 1 that contains the variable and outcome definitions.

Most importantly, no measurements or events (including hospitalizations) that occurred after the beginning of the COVID-19 pandemic in Israel were included as predictors. As the reviewer correctly states, that would constitute a severe data leak. In order to make sure that this central point is clear, we added this comment to Table 1: "*All variables were extracted at or before February 1st, 2020, prior to the onset of the COVID-19 pandemic in Israel*". In addition, the Method section states that "*The final baseline model was applied to the entire CHS population over the age of 10 years as of an index date of February 1st, 2020, which was chosen to avoid the effects of COVID-19 (the first COVID-19 case was identified in Israel on January 29th, 2020)*".

-In the version with track changes, Figure 4 has not changed. The calibration and decision curve must surely look different now that more data was added. Please check that the correct versions are used.

It seems the software's version-comparison feature failed to tag the change. The figure that was included in the resubmission file was changed, although the figures were similar in their overall structure. For the convenience of comparing the changes through all the resubmissions (because they are indeed subtle), we include all three versions side by side:

Ben Van Calster, KU Leuven, Belgium

Reviewer #2 (Remarks to the Author):

The authors have satisfactorily addressed all of my comments.